# Intercultural Education for Sustainability in the Educational Interventions Targeting the Roma Student: A Systematic Review

Norma Salgado-Orellana, Emilio Berrocal de Luna * and Christian Alexis Sánchez-Núñez

Department of Research Methods and Diagnostic in Education, University of Granada, 18071 Granada, Spain; nsalgado@correo.ugr.es (N.S.-O.); cas@ugr.es (C.A.S.-N.)
* Correspondence: emiliobl@ugr.es; Tel.: +34-958-249-847

**Abstract:** The ethnic and cultural diversity of today's society demands an intercultural education that promotes equal opportunities and the inclusion of minority groups at risk of social exclusion. Various policies and governmental strategies are directed at favoring social inclusion and reducing situations of discrimination and exclusion. One of said ethnic minorities is the Roma community. This article talks about measures for reversing the high rates of absenteeism, dropout, and school failure. The aim of this systematic review in the educational context is to analyze programs and interventions that have been made to promote the educational inclusion of Roma students. Three databases were used—Scopus, Web of Science, and Eric—to examine 419 articles. After a selection, based on an inclusion criterion, which follows the guidelines given by the Declaration of Preferred Reporting Items for Systematic Reviews and Meta-Analyzes (PRISMA), 17 articles were chosen for the analysis. The main findings shed light on data for programs and interventions developed mostly within the school setting and for students of elementary and secondary education, however, there are also interventions that are developed in the community and extracurricular context. The works are centralized in valuing the cultural diversity of the Roma people with the Roma students but not just exclusive to them. Highlighted is the implementation of teacher assistants and/or intercultural mediators from the Roma community to be role models or references for the students, and family education participation is considered common practice. Such characteristics in educational interventions contribute to an inclusive education in an intercultural and sustainable manner.

**Keywords:** intercultural education; educational interventions; Roma student; systematic review; PRISMA

## 1. Introduction

Considering the importance of intercultural education as a priority for sustainable social inclusion in the European community [1,2], this study has attempted to identify, through a comprehensive literature review, programs aimed at Roma student interventions within an educational context, which have developed and promoted changes at the individual and the group level, this being one of the guidelines of intercultural education for sustainability, and resolutions for situations of exclusion and marginalization of students that improve their quality of life [3].

An increasingly ethnically and culturally diverse [4] society requires intercultural education to be a strategic resource "towards sustainability, equity, peace, and social cohesion" [5] (p. 123). In this context, intercultural education programs should be designed and implemented with the goal of sustainability [1,5], and maintained to guarantee the continuity of intervention programs aimed at supporting social inclusion of the Roma people [6].

A couple of decades ago, there were concerns about the truancy problems of Roma students, and the high levels of school failure. In this regard, Abajo [7] calls for a profound change, at the social level, to ensure a quality education that enables them to develop as citizens in their own right. Faced with these facts and the publication of official reports on the education of the Roma students [8], this is becoming a priority issue for public entities and European governments.

In this way, one of the priorities at the European level is social integration, as shown by the Europe 2020 Strategy [2], especially aimed at those most disadvantaged and at risk of social exclusion. The member states of the European Union have designed efforts to overcome early school dropout by applying different strategies in order to achieve their goal Europe 2020. For the full and successful social integration of Roma students, educational success is necessary [9] and to achieve this, member countries of the European Union are committed to a set of strategies aimed at inclusive growth, boosting school attendance rates and the effective transition to higher education. [10].

It is true that education has made significant progress; however, the percentage of Roma students leaving the educational system is 24 times higher than the whole population [10], with the Roma population also displaying higher dropout rates [11].

For Antunez, Perez-Herrero, Nunez, Burguera, and Rosario [12], the decade of Roma inclusion (2005–2015) has been firstly, a step towards social inclusion, an opportunity for collaboration between European countries, and the recognition of different strategies and actions that promote equitable access to education. To this regard, the Open Society Institute [13] has 10 recommendations related to equitable access to education for Roma, where it mentions the implementation of programs aimed at this group ("regularly review, improve, and implement policies and programs for Roma"(p. 5)).

For Kilicoglu and Kilicoglu [14], the educational situation for Roma students is similar in European countries, which following the guidelines of the European Union have implemented educational programs and best practices [12,15] in schools.

One of the recommendations is to transform schools into learning communities [16], and for this they have established different programs aimed at Roma students, which seek to involve and engage the entire community in order to achieve a quality education and encourage Roma students to pursue secondary studies [9]. In this regard, partnerships play a significant role in educational inclusion and have to be sustainable so that they can provide the "opportunity to learn the rights of citizenship" [1] (p. 139). However, they should be implemented along with other structural measures and strategic interventions [17].

As an example, Antúnez et al. [12] point out various initiatives and good practices developed in Europe to improve access and school performance of Roma students. They identified personal factors that promote academic achievement which should be encouraged and promoted by educational intervention programs [18] which help to improve the opportunities for school achievement of Roma students. Another example, from Spain, is the implementation of educational laws, which have followed European guidelines concerning the implementation of intercultural education programs, promoting and spreading best practices and improving teacher training [19,20].

Intercultural education is a perspective that is based on cases of "equity", "diversity", "inclusion" and "democracy". It is characterized by the search for equality, the formation of citizens, and the respect and positive valuation of cultural diversity [21], as well as the establishment of common values [22,23] through a critical dialogue. Sales and Garcia [23] propose that active citizen participation and criticism are essential to develop a democratic society based on equality, tolerance, and solidarity. Intercultural education proposes that it is vital to work in coexistence recognizing universal values in difference while constructing broader and richer cultures. In this line, it is necessary to construct a more interactive society, where its culturally diverse citizens learn to live together and thus contribute to improving and developing socially, especially through the education of the youngest, by turning it into an education for each and every member of society [24]. It is a model of holistic pedagogical intervention, composed of all those dimensions that make up the integral development of apprentices. For some, intercultural education should be oriented towards a global educational project, education for

intercultural citizenship, which requires a critical formation, participatory model based around common issues [25]. Intercultural education needs an inclusive educational commitment from socio-educational institutions with the community, an education built within the framework of the relationships between the agents (social and educational) to be established, an intervention model that responds to the need to live together and grow in an intercultural society and that goes beyond focusing on the school and goes to the community to achieve the following goals [22,26,27]: (1) awareness and training of citizens; (2) acceptance, recognition, and a search for common values; (3) construction of broad and shared cultures; (4) integration and social, educational and labour insertion; (5) democratic and participatory management of the centres; (6) social and educational centres involved in the community and vice versa; and (7) dialogue, meeting, coexistence, cooperation and the overcoming of conflicts.

Therefore, intercultural education is a valid model to contribute to social and human sustainability; it helps to maintain and increase social cohesion, respect for cultural heritage, and guarantee the fulfilment of cultural rights. That is, to obtain sustainable development, intercultural education contributes to the defence of cultural diversity and intercultural miscegenation as the heritage of all humanity and helps to combat all kinds of racism and ethnic or social barriers contributing to the equity of genders and races among the people who live in our towns and cities; societies that are increasingly culturally diverse and whose future generations of citizens are the children who today learn to live in our schools.

Furthermore, the classical Byram proposals on intercultural citizenship (taken from Byram, Perugini and Wagner [20]), which defend the notion that intercultural learning is the knowledge and also the understanding of the people with whom it interacts, that learning should include attitudes and values related to critical cultural awareness and social justice and ultimately that learning is achieved in interaction with others, hence emphasizing the importance of learning both in and out-of-school and community environments, respectively.

Consequently, in order to improve the social inclusion of the Roma population, it is important to ensure that children and young people have a quality education, improving their school level [12], which would increase their employability [16] and at the same time position intercultural education as a corner piece of sustainability in today's society, thus keeping a balance between knowledge, learning and social equity whilst maintaining equality between sustainability and intercultural education programs [1].

According to the above, the objective of this study is to analyze in a descriptive way the interventions directed to the Roma students in the literature, within the framework of intercultural education that have been carried out in the educational context.

In order to fulfill the aforementioned objective, the following specific objectives have been established:

- Identify the educational context where the intervention has been carried out (school, extracurricular or extracurricular, community or in more than one context).
- Describe the main characteristics of the interventions carried out with Roma students according to the purpose of the intervention, school level, participants and results.
- Assess the implications of the results of the interventions for Roma students.

The investigation's questions relate to: What programs or interventions were carried out with Roma students? What are the educational levels of the programs or the interventions that have been carried out? What are the settings of these programs or interventions? What have been the results of interventions and their implications for Roma students? Do intervention programs respond to the political guidelines on educational inclusion of Roma?

Finally, the following study corresponds to a systematic review, which has followed the guidelines provided by the PRISMA Declaration [28] to analyze the literature and identify the research that answers the questions in this study, following a well-defined, orderly methodology and objective. Its main aspects are presented in the following section.

This descriptive analysis, through a systematic review of previous studies, will identify common and successful aspects (as well as risks and limitations) that, within the framework of intercultural education, present the educational interventions carried out with the group of Roma children for their social and educational inclusion. For this reason, it is important to identify, systematize and share the main characteristics, while trying to contribute to the generation of useful knowledge for professionals, educators and researchers in this field of intercultural education; a knowledge that allows them to make decisions when carrying out future programs.

## 2. Methods

The method used in this study was a systematic review, following the guidelines provided by PRISMA Declaration [28] according to the flowchart of four phases in addition to its checklist and report items.

### 2.1. Selection Procedure and Datamining

The literature search was started in December 2018 and carried out in three data bases, Scopus, Web of Science, and Eric, including articles published up until February 2019. The search did not have a limit to the year of publication, that is to say, any article before February 2019 was considered. In each database, a search equation was carried out according to each procedure for the advanced search, using keywords like "gypsy student", "roma student", "traveler student" gypsi * (together with the Boolean operator OR). Also, the words "education programs" "education intervention" (together with the Boolean OR), in general the equation was structured as follows: (gypsi * OR "gypsy student" OR "roma student" OR "traveler student") AND ("education" OR "education programs" OR "education intervention"). The results were included in a computerized data matrix made with Microsoft Excel—the statistical package of Office 2016 for MacOS—then duplicates were eliminated. In addition, we used the reference manager EndNote X8 (Thomson Reuters Corporation), which also facilitated the identification of duplicates and the management of bibliographic references used in the study.

The researchers followed the PRISMA guidelines [28] for data extraction through their 27 items checklist. The first two authors made the independent selection of each study, by reading the title, abstract and keywords (419 investigations), and the third author resolved disagreements.

In the process of analyzing the data from the articles, the three authors worked independently. Then the extracted data was summarized based on the research questions, regarding: (a) reference article, (b) main background, (c) study objectives, (d) beneficiary, (e) context, (f) school level and (g) main results. This analysis was reviewed and analyzed by the two main authors, obtaining a 90% agreement between them with discrepancies resolved collaboratively among the three authors. The results are presented in Table 1.

Table 1 presents the sample according to each database, according to the inclusion criteria.

**Table 1.** Number of items (and percentage) according inclusion criteria.

| Search Criteria | Scopus | WOS | ERIC | Duplicates | Total |
|---|---|---|---|---|---|
| Initial search | 399 (50.95) | 283 (36.14) | 101 (12.89) | | 783 (100) |
| Inclusion criteria (1) * | 216 (42.1) | 229 (44.63) | 68 (13.25) | | 513 (100) |
| Inclusion criteria (2) ** | 28 (40.57) | 25 (36.23) | 10 (14.49) | 6 (8.69) | 69 (100) |
| Inclusion criteria (3) *** | 2 (11.76) | 8 (47.05) | 1 (5.88) | 6 (35.29) | 17 (100) |

* Criterion 1: articles written in English or Spanish areas of social sciences and psychology. ** Criterion 2: theoretical and empirical studies whose titles, abstracts or keywords indicate intervention programs aimed at Roma students. *** Criterion 3: the focus of the study should be to present educational programs or interventions aimed at Roma students.

## 2.2. Criteria for Inclusion of Articles

It is worth mentioning that there was process deduplication, which was conducted during analysis inclusion in Criterion 2. Table 2 shows the amount and percentage of duplicates in all databases items in the selection of items.

**Table 2.** Number of duplicate items and (percentage).

| Database | Scopus-Eric | Scopus-WOS | Wos-ERIC | Scopus-Wos-ERIC | Total |
|---|---|---|---|---|---|
| Duplicate items | 13 (13) | 58 (61.7) | 3 (3.19) | 20 (21.27) | 94 (100) |

## 3. Results

The first search of the three databases used in this study delivered a total of 783 studies and according to the first criteria of inclusion, only 513 items were considered, from which 94 of these were duplicates. Therefore, the reading of titles, abstracts, and keywords was performed on 419 items. At the stage of screening, 69 articles were selected from which 21 were excluded for not meeting Criterion 2. The number of items evaluated for eligibility was 48, from which 16 were removed, it should be noted that two of these articles could not be found. In assessing the quality, all three authors participated in evaluating the 32 studies. Finally, 17 met the inclusion criteria and were selected for qualitative synthesis.

In the process of analyzing data items, the three authors participated independently. The articles selected for analysis were summarized based on the research questions, in relation to: (a) article reference, (b) main background, (c) study objectives, (d) beneficiary, (e) context, (f) school level and (g) main results. This analysis was reviewed and analyzed by the two main authors, obtaining a 90% concordance between these, the discrepancies were resolved collaboratively between the three authors. The articles included in the qualitative synthesis and its main background have been presented in Appendix A.

According to the flow diagram of the PRISMA declaration [28], Figure 1 is presented showing the flow of information followed in this study, for the search and selection of articles.

Then, a narrative synthesis is presented to report the results and the findings 153 from the analysis of the 17 items.

For the organization of the analysis, the items have been classified by whether the program or intervention was performed at school, or carried outside of the school hours, it is to say, extracurricular or in the community, or both (school, extracurricular and/or community).

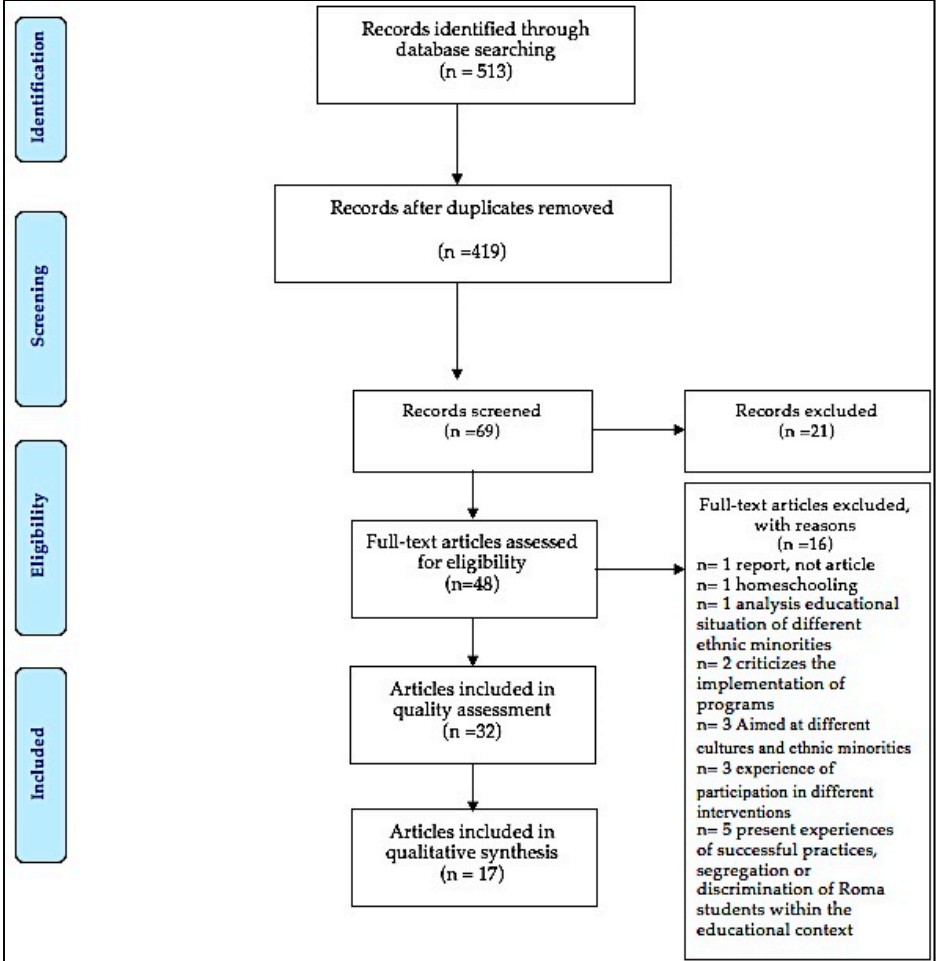

**Figure 1.** Flow diagram depicting the search and selection of studies process according to the Preferred Reporting Items for Systematic Reviews and Meta-Analyzes (PRISMA) Declaration.

### 3.1. Studies Carried out at School

Of the total articles (n = 17) ten of them have been carried out at school [3,29–37].

These programs or interventions have been directed mainly towards Roma students in secondary education [30,31,34]. The first two took place in Spain and the third in Greece. Within these, [30,31] serve a similar purpose, in that they both seek to promote inclusive activities to promote the educational inclusion of Roma students at their school center, improving the relations, and appreciation towards Roma culture. Study [34] presents a project that seeks to involve Roma students in the development of their knowledge, using new technology; additionally, it also promotes a positive sense of identity and Gypsy culture. The three papers show improvements in behavior, attitudes, valuation and arrangement of schoolwork. Additionally, [34] shows the importance of using new digital media to help Roma students overcome barriers that they encounter in the educational system.

In primary education, there are three articles [32,35,36] conducted respectively in Serbia, Hungary and Turkey. The three articles present educational interventions with innovative and motivating methodologies. The first [32] uses a teaching laboratory experiment, which aims to increase the quality and quantity of knowledge, through motivating experiences in the laboratory, related to ecology and environmental care. The second, [35] is carried out in different grades of primary schools. Interdisciplinary programs are carried out using technology, communications, and informatics to motivate, develop skills and promote the in-school achievement of Roma students. Finally, the last article, [36], promotes music education, through the creation of a choir of Roma students, to strengthen aspects of socialization and skills related to music. The three articles show favorable results, according

to their objectives, highlighting the importance of the use of teaching methods, the promotion of Roma culture through music, and the importance of social and cultural aspects [36].

Likewise, some articles have developed interventions or educational programs for two levels, mostly aimed at students of elementary and secondary education [29,33,37] and all set in Spain. The first, developed a nutrition education program, designed as a community intervention in the school context with the intention of promoting healthy eating habits and self-care. Flecha and Soler [33] conducted a case study in the school context which also involves the entire educational community and family. Their intervention is part of the integrated project INCLUD-ED. It was carried out in a school that was identified to be in a critical situation, in which they implemented a dialogical inclusion contract. They carried out successful educational actions to transform the educational and social context, based on the dialogue [38]. Activities such as family involvement in decision-making, interactive groups, assemblies, and Roma volunteers in the classroom were encouraged. Aparicio Gervás and León Guerrero [37] conducted the "Increscendo" project, funded by the Symphonic Orchestra of Castilla y León. It was based on musical training and successful experiences of training choirs in other parts of the world. It is an intercultural and community project, carried out in a public school (Antonio Allúe Morer). It aims to facilitate strategies and promote the educational inclusion of Roma children with the creation of the choir, where students and teachers participate, and it also addresses emotional, affective, cognitive and social aspects of students. Generally speaking, the three studies presented favorable results, achieving the expected results in the direction of their goals. In the case of Pérez-Rodrigo and Aranceta [29], after two years of implementation of the program on nutrition education, the development of skills and knowledge delivered in the program like self-care behaviors is evidenced; however dental hygiene did not meet the expected results, despite their having given the knowledge, skills and equipment to complete the tasks. Additionally, family involvement was intermittent at meetings, despite worrying about sending the materials to carry out the cooking workshops. In the case of [33], the School La Paz developed good practices through four years of project implementation, which has contributed to the educational achievement of all students in the school, evidenced concretely by the results of standardized external evaluations. In addition to other achievements, it increased educational opportunities for secondary education, which caused young people to continue their secondary education, increased school attendance, and significantly reduced truancy. Finally, the results considered in [37] show that the choir was able to develop attitudes and values in line with intercultural education, where music played a central role; their work is projected in Spain and in other international contexts.

Among the programs and interventions carried out in the elementary school setting, only one article was identified for the levels of preschool and primary education, [3], which was conducted in Latvia. The project follows intercultural education for sustainability by focusing on an inclusive school in order to improve the quality of life of the educational community. "The Roma Teaching Assistant Program" considered the implementation of multicultural classrooms in nine different schools with teaching materials and resources for teaching and learning related to culture, language, traditions, and the history of the Roma people, by which some of these materials were made by families in parenting workshops. These cultural elements should be present daily in the inclusive classroom. In addition, support centers were set up for Roma families and teaching assistants belonging to the Roma, who worked collaboratively with teachers and Gypsy parents, which they recognized as a great support. Through these strategies, the intention is to stimulate the inclusion and the adaptation of Roma children to school. Their results showed a better adaptation to school while improving their social skills, a general satisfaction from the families for seeing their ethnic necessities considered, and finally this inclusive model meets sustainability for intercultural education.

*3.2. Studies Carried out in the Extracurricular Context*

There are two programs or interventions that have been carried out in the extracurricular context, [39,40]. They are considered out of school because their activities happened after school hours.

Firstly, Messing [39] presents a case study, the "Learney" project, and secondly, Rosario et al. [40] present an empirical study; however, the former was extracurricular. In the case of Messing [39], a space was implemented where students could go after school to have fun and complete their homework and [40] the intervention program used narration as the main tool (Sarilhos do Amarelo [41]). The aim of Messing [39], besides providing a space, was to encourage the continuation onto secondary education while strengthening personality, identity, cognitive skills, community building, educational and vocational orientation. The program [40] developed different processes of self-regulated learning and behavioral commitment, using microanalysis methodology and establishing two groups (control and experimental: 18 children, 17 children, respectively). Finally, the first paper [39] does not present empirical results, only limitations to the success of the project and conclusions. The second study [40] showed that students in the experimental group increased in measurements of the dependent variables after the second evaluation, the measurements of the control group remained the same in all three evaluations. In order to see the effects of the program, the dependent variables were analyzed in three stages. When considering the two dependent variables simultaneously, differences between groups' measurements for 2–3 and 1–3 were found. For the behavioral commitment variable, significant differences between the two groups were found according to the moment of measurement, as occurred with the cognitive engagement variable, with the greatest differences occurring between 2–3 and 1–3. The results of the control and experimental groups demonstrate the effectiveness of the program in encouraging school engagement.

### 3.3. Studies Carried out in the Community Context

There are two interventions carried out in the community context, that is, the community, the neighborhood or the place where the Gypsy students live [42,43].

In the first one, Rosario et al. [42] developed an intervention program, "Knock, knock It's time to learn!" with a longitudinal design (4 years), and the second study, by Bhabha et al. [43], conducted a program, "Reclaiming Adolescence", using the design of youth participatory action research. The two studies carried out interventions in a community setting, and the former [42] considers the need to investigate the relationship between social networks, community, neighborhood and behavioral commitment, and in the case of [43], documents the opinions of Roma youth about their experiences with discrimination in secondary and university education, situating the youth as researchers. In accordance with their objectives, Rosario et al. [42] promote behavioral commitment and school performance. The program specifically addressed the behavioral commitment (truancy and classroom behavior), and school performance (math course and progression of each student). The project [43] focused on three objectives: (a) obtain background on educational and professional opportunities; (b) train 20 young (11 Gypsies and nine non-Roma, from 15 to 24 years) in research skills; and (c) develop research strategies suggested by the young researchers; however, their main goal was that the young Roma participated in all stages of the investigation. Both projects were developed differently, but both targeted the community level. In Rosario et al. [42], every morning for four years, Roma research assistants would knock on the doors of the experimental group, inviting the children to go to school, and then together they walked to school. In the project [43], the youth designed interview questions for their companions (Gypsies 163, 117 non-Roma), parents, and representatives of institutions. Among the findings, both studies reported favorable results, the first [42] obtained measurements for four dependent variables: (a) lack of assistance; (b) classroom behavior; (c) grades in math, and (d) school progress. Generally, children in the experimental group improved in the four variables assessed, and additionally, the neighborhood cooperation helped with the participation of the children. This strategy proved to be effective in helping Roma children without devaluing their own customs. In the case of [43], there were differences in opinion between Roma parents, non-Roma parents, and state representatives, that would account for a level of disagreement and bias against the Roma. A central aspect of this investigation was the impact of discriminatory situations in educational and professional expectations. However more impactful was the Roma youth's experience

with discrimination, as they were less likely to perceive education as an important part of their future. That is to say, discrimination has an effect on an individual's confidence and educational expectations; however, they perceive equal opportunities at the social level.

*3.4. Studies Conducted in More Than One Context*

Three articles were found that developed in more than one context at the same time [44–46] either school, community, and/or extracurricular.

The first two articles correspond to the project Service Training for Roma inclusion (INSETRom) of IN-Service [44,45]; however, both were developed in different countries, Austria and Cyprus. The third, Battaglia and Lebedinski [46] carried out the intervention program Education for All, coordinated by the Ministry of Education of Serbia.

All three projects were carried out in primary schools, but only [44,45] also included high school, only [44] a special school, only [45] a rural primary school, and only [46] at a preschool. The two projects [44,45] included actions during school hours and extracurricular hours, and in [46] the program took place in three contexts, school, out of school, and within the community. The three projects had the help of a Roma assistant who served as a mediator between the families and the school and delivered educational support to the children after school. Particularly, program [46] asked the pedagogical assistants (one per school) that they participated and helped in the classes of Roma students, and visited the families and community once a week.

Programs [44,45] aimed at improving the effectiveness of education for the Roma students, while strengthening relationships between the family and the school, including three phases (the evaluation of the needs of the whole educational community, teacher training, and teacher intervention in the classroom). The two studies mention the use of semi-structured interviews in the first phase directed to different members in the educational community, and study [44] gathered participant observation in the extracurricular educational program. In the case of [46], the objective was to assess the impact of the program during the first year in three aspects: dropouts, school attendance, and school performance. The study measured the impact of the program comparing school's early-enrollment (September 2009) to school's late-enrollment (November 2010). The differences in the findings are notable, in project [44] it is evident there are contradictory aspects related with school attendance and educational achievement between teachers and parents. In terms of the behavior of children and social inclusion, teachers, Roma students, and their families signal evidence to adequate inclusion in the classroom and with their peers. In these schools they seemed not to have problems with marginalization, since, according to their teachers the diversity of their classes helped to avoid these situations. Regarding the teaching process, teachers claimed that there were no differences in teaching based on ethnicity, only accordingly to each student's own abilities, however, they felt ill prepared in the teaching of ethnically diverse groups. The relationship between teachers and Roma parents was strengthened through the mediation of the Roma assistant. The results of teacher training showed changes in classroom practices, improving the skills in working with Roma students. The findings of the project [45] bring to light irregular attendance at school, according to their teachers, despite evidence of an improvement over time. Teachers noted difficulties of the Roma children in the knowledge of the official school language (Greek), a gap between the teacher and family relationship is also considered. The teachers pointed out the social inclusion of the Roma children in school was different, evidencing situations of prejudice in the form of bullying. The results from the teacher training revealed the need for more training on practical issues in teaching Roma students. Parents noted concerns regarding bullying, language difficulties, cultural problems, and isolation. The children wanted the teachers to know more about their culture. Overall, the training program did not address the concerns of teachers regarding the actual teaching of Roma children. Among the results of the study [46], the hours of absence reduced, the performance in the subjects of Serbian and mathematics improved, a greater impact from the program in schools with fewer Roma children was observed, and it showed that Roma

teaching assistants are a valuable contribution in teaching Roma students because they know their culture and are accepted within their community.

*3.5. Summary of Studies*

To answer the questions of this study a summary is presented in Table 3. First off, it is evident that there are more programs or interventions that are performed in one context (school) 58.82% than there are of programs or interventions that consider more than one context (17.64%). Regarding the educational level, a greater quantity (nine articles) employ interventions in more than one educational level, of these the most [29,33,37,39,45] are concentrated in the elementary and secondary levels; two articles, [3,46], focus on preschool and elementary education; one article [44] worked with elementary, secondary, and special education, and one paper [43] presents a program in secondary and higher education. A higher percentage of programs or interventions involved not only the Roma community (70.58%) while 29.41% were exclusively aimed at Roma students. Finally, Spain is the country with the most programs or interventions found for this study.

**Table 3.** Summary of studies.

|  | Quantity | Percentage |
|---|---|---|
| **Context** | | |
| School | 10 | 58.82 |
| Extracurricular | 2 | 11.76 |
| Community | 2 | 11.76 |
| More than one context | 3 | 17.64 |
| **Total** | 17 | 100 |
| **School level** | | |
| More than one level | 9 | 52.94 |
| Primary | 5 | 29.41 |
| Secondary | 3 | 17.64 |
| **Total** | 17 | 100 |
| **Recipients** | | |
| Only Roma students | 5 | 29.41 |
| Roma and non-Roma students | 12 | 70.58 |
| **Total** | 17 | 100 |
| **Country where the program or procedure is performed** | | |
| Spain | 5 | 29.41 |
| Serbia | 3 | 17.64 |
| Portugal | 2 | 11.76 |
| Hungary | 2 | 11.76 |
| Cyprus | 1 | 5.88 |
| Greece | 1 | 5.88 |
| Austria | 1 | 5.88 |
| Latvia | 1 | 5.88 |
| Turkey | 1 | 5.88 |
| **Total** | 17 | 100 |

## 4. Discussion and Conclusions

Based on this systematic revision and the results, there has been a profound understanding of what has been reported related to the development of the educational intervention directed to the Roma student body, within an intercultural education margin for sustainability. The objectives of the different interventions relate to the intercultural educational plan. They coincide with Ladson-Billings' [47] beliefs, who defines three criticisms which cultural education is based on: the student academic success, cultural identity and development of critical thinking in empowering their ethnic group.

There is equally a contrast between Jiménez Gámez and Goenechea [24], Muñoz Sedano [26] and Jordán [22]. At the same time, the programs respond to the objectives of sustainability, taking into consideration that education should foster abilities and knowledge to produce solutions in future sustainable societies [48].

It is important to distinguish that the interventions identified based on this systematic revision cover educational levels from early childhood up to secondary education. Only one program developed within a community included young adults from secondary education and university [43]. There clearly still exists a hesitation to implement these ideas in the first years of education [40]. It is proven that these interventions do not coincide with the educational politics in reference to the young Roma participation in universities [16]. According to Matache [49], there have been few actions and there has been little funding to guarantee young Roma access to universities. The data contributed by Brüggemann [50] agree that in the countries with high Roma population, 90% of the young Roma are excluded from secondary education and close to 99% from universities. According to The Open Society Institute [51], issues of inequality with access to education increase within the level of education, the difficulties and deprivation of opportunities that are related with political factures, social economics and historical [43].

In relation to what was proposed by the European Commission [2], the use of innovative methodologies to favor the educational inclusion of Roma students is noted. These are alternative strategies aimed at increasing motivation, cognitive abilities and school achievement, which are adequate to get an equitable access to education and teaching and learning based on their culture [34,35]. So, it is verified that elements of the culture, language, traditions and history of Roma are included in the classroom setting, in the teaching-learning process [3] and in the out-of-school surrounding, which contributes to an improvement in the assessment of the Roma culture by the community according to intercultural positions [25,27].

Open participation and ongoing dialogue with the Roma community would be in direct relation with what was proposed by Greenfields and Ryder [52] about the meaning of making studies "with" and "for" Roma people instead of "in" them. The importance of encouraging the participation of the family, community and environment is emphasized, and [44] points out that aspects related to Intercultural Education could be addressed not only through programs within schools, but also through bodies that involve the Roma communities. In this sense, Intercultural Education and dialogue can provide an effective response and solutions to the different challenges that emerge from a culturally diverse society [38,53]. Therefore, those interventions developed in the community context where schools and communities collaborate; Greenfields and Home [54] call it community participation, are relevant and successful. The family has an important role as a source of knowledge and transmission of the culture, values and the Roma tradition hence the governmental programs must go in hand in hand with the local realities of the Roma community, as indicated by Marc and Bercus [55]. In this sense, the analyzed interventions have focused on valuing the Roma culture, establishing cooperation strategies and family and community participation, establishing an open dialogue between cultures, through cooperation, interaction and interrelation for the formation of a fairer coexistence, as affirmed by Araque [30].

In several programs interventions with the participation of students, teachers, Roma assistants, families, Roma communities and neighborhoods have been considered, endorsing the success of the named "communities of practice" or "learning communities" since it is a principle of intercultural citizenship that students interact with people outside the classroom [20,25]. In this way, the importance of the "teaching assistant" leading figure has been identified. It is important to implement educational interventions where the referent or assistant is a person of the same ethnic group and not of the majority culture; it promotes a feeling of cultural belonging in children and young Roma and of equality of cultural status [21,23]. It also facilitates the understanding and connection between the school and the child, being a great support for all the students of a class [3,46].

The interventions developed after school schedule would help children and young people to participate in activities that facilitate their educational and social inclusion. According to Messing [39],

it can be effective when there are a high index of poverty, social and ethnic segregation. As Byram, Perugini and Wagner [20] point out, it consisted of "breaking through the classroom walls, into the society in which students live; they extended their inquiries into the social world of parents and friends" (p.22). It is considered that the implementation of an out of class program is a difficult and demanding task, considering the cultural aspects of the Roma community [40]. This would be in direct relation with that mentioned by Battaglia and Lebedinski [46], who propose that in the United States there has been thorough evaluation of school programs aimed at underprivileged groups, which include extracurricular programs.

A fundamental aspect in intercultural education is teacher training [56]. Teachers working in these contexts need a constant update, cooperation across the curriculum and intercultural competence [38,57] to develop strategies and methodologies appropriate to the cultural characteristics of this group. In this sense Biasutti, Concina and Frate [58] point out that through training instances teachers encourage educational inclusion and the development of students from a sustainable social point of view, because the social sphere is one of the pillars of sustainability.

Therefore, for the development of an intercultural education for sustainability, interventions that favor knowledge, cooperation and interrelation between different cultures are recommended. These could take the form of social and educational actions in environments of cultural diversity directed and participated by students, both of minorities as well as cultural majorities.

All this is in accordance with Sustainable Development Goals for 2030 [59], specifically in that it aims to ensure responsive, inclusive, participatory and representative decision-making at all levels (goal 16.7) and in working to the achievement of goal number 4, which is maintains that obtaining a quality education is the foundation to creating sustainable development. Goal number 4 aims to ensure that all girls and boys complete free, equitable and quality primary and secondary education leading to relevant and effective learning outcomes (4.1); that all girls and boys have access to quality early childhood development, care and preprimary education so that they are ready for primary education (4.2), ensure equal access for all women and men to affordable and quality technical, vocational and tertiary education, including university (4.3), ensure that all learners acquire the knowledge and skills needed to promote sustainable development, including, among others, through education for sustainable development and sustainable lifestyles, human rights, gender equality, promotion of a culture of peace and non-violence, global citizenship and appreciation of cultural diversity and of culture's contribution to sustainable development (4.7).

It would also be convenient to favor the participation of the Roma women in the different educational interventions, either as a teaching assistant or as an intercultural mediator in schools and communities, giving them greater representation in line with goal 5, which is to achieve gender equality and empower all women and girls [59].

Finally, given the lack of longitudinal studies on educational interventions that shed light on the success or academic achievement of the participating Roma students, it is recommended to conduct evaluative studies that support effective educational practices [57]. The democratic value in educational interventions is also considered very important, with the participation of the Roma community, social, educational and research institutions in their design to achieve true social and educational inclusion [42,60].

## 5. Limitations

Like any systematic review, this study has some limitations. The first is related to the language of the articles, only studies written in Spanish and English were selected. Similarly, despite having conducted an exhaustive literature review, it has its limitations based on the search terms in each data base; however, a clear and systematic overview of the results delivered by each author and their findings has been delivered. Another limitation related to the databases is, that despite using the most recognized and used data bases in education, future studies should consider expanding the search to more databases.

An important limitation is the small number of articles included in this study, which have been very difficult to reach, since the literature often only reports particular educational experiences presented through interviews or qualitative studies, with few studies focused on evaluative programs or educational interventions.

Finally, it is considered as a limitation in the search, selection and analysis of research that we only included articles and did not consider other types of documents, which perhaps could have provided further information relevant to the educational intervention towards Roma students.

**Author Contributions:** N.S.-O. and E.B.L. were responsible for study statement and literature search in data base. N.S.-O., E.B.L. and C.A.S.-N. participated in the review of articles and the quality evaluation of these. Also, the three authors participated redaction and manuscript revision being more important the participation of N.S.-O. and C.A.S.-N. in the elaboration of the introduction. N.S.-O. and E.B.L. in the method, process and the results. The three authors were responsible for discussion, conclusions and limitations.

**Funding:** This study was funded by CONICYT Scholarship PFCHA 72190263. Also has the support of the Research, "Innovation and Improvement in Andalusia" (HUM 126). Access to databases has been done with the Virtual Private Network (VPN) from the University of Granada.

**Acknowledgments:** Thanks to David G. Sachs, graguated by the University of New Mexico (EE.UU.) and posgraduate by University of Granada (Spain), and Marcela Bastías, Spanish tutor at the Global Enagagement Centre, University of Coventry, Conventry, UK for the help given in the translation of this article.

**Conflicts of Interest:** The authors declare no conflict of interest. The funders had no role in the design of the study; in the collection, analyses, or interpretation of data; in the writing of the manuscript, or in the decision to publish the results.

## Appendix A

**Table A1.** Main aspects of the items considered in the systematic review.

| Reference | [29] |
| --- | --- |
| Background | Pilot program in nutrition education, designed as a community intervention. It was developed in the framework of social learning theory and self-empowerment. Three ways considered action: class, a workshop and cafeteria, as well as a plan for the families of students. |
| Objectives | To promote healthy eating habits, along with the development of skills and self-empowerment |
| Beneficiary | 150 children (8–12 years, 80% Roma) identified a school in a deprived area of Bilbao, Spain. |
| Context | School |
| School level | 3rd–6th Primary E.; 1st Secondary E. |
| Main results | After two years, the program evaluation showed improvements in knowledge, skills and behavior of children. The qualitative evaluation showed a positive attitude in the program. The involvement of families in the meetings was low, dental hygiene showed little improvement. |
| Reference | [39] |
| Background | Learney school program that promotes inclusion, addressing underachievement and high dropout rates. It organizes activities and helps children to finish Primary education. Various teaching methods are identified. |
| Objectives | To provide time and space for Roma children to carry out after school activities. |
| Beneficiary | Children and youth (6 to 18 years) in Hungary |
| Context | Extracurricular |
| School level | Primary E. and Secondary E. |
| Main results | The case study shows, no results of interventions. |
| Reference | [30] |
| Background | Intercultural Education experience to solve problems of racism and relationships experienced by young Roma when attempting to integrate school. |

**Table A1.** *Cont.*

| | |
|---|---|
| **Objectives** | **To develop** tolerance and reduce prejudice towards Roma culture. To create an environment that fosters relationships of respect. |
| **Beneficiary** | 16 students (17–23 years, 25% Gypsies), the course of social guarantee of IES, Orcasitas area, Madrid, Spain |
| **Context** | Classroom-workshop. |
| **School level** | Secondary E. |
| **Main results** | The interviews showed a change in valuation among students. Questionnaires are increased in the provision for work and respect for Roma culture. Positive change of attitudes, (classwork and relationship with peers) are witnessed |
| **Reference** | [44] |
| **Background** | The article presents the three main phases of the In-Service Training for Roma Inclusion (INSETRom) Project. It discloses the main activities and results of the three phases proposed in the project (needs assessment, curriculum development and teacher training, implementation and evaluation of interventions) |
| **Objectives** | To promote educational inclusion, improving the effectiveness of teachers in their schooling, strengthening the relationship between community and school. |
| **Beneficiary** | Roma students (7–10%), between 5 and 14 years, teachers and parents from three schools (primary, secondary and special) in Vienna, Austria. |
| **Context** | College Extracurricular |
| **School level** | Primary E., Secondary E. and Special E. |
| **Main results** | The results of the teaching are presented. They indicate that in the implementation phase training has led to changes in practices and teaching methods, improving skills in the daily work with Roma students, and a better relationship between family and school is evident. |
| **Reference** | [45] |
| **Background** | The article presents the results of EU-funded INSETR It focuses on the needs assessment phase of the entire community, and on the implementation of the training program for teachers. |
| **Objectives** | To promote inclusive education in schools in Cyprus and to identify the needs of the educational community for interventions. |
| **Beneficiary** | Three schools (urban primary, secondary and rural primary) with 62 Roma students in total, in Cyprus. |
| **Context** | College Extracurricular |
| **School level** | Primary E. and Secondary E. |
| **Main results** | Teachers showed pessimism about the results of training and implementation in teaching Roma students. The results show that educational inclusion strategies should be comprehensive, and also that there should be a greater focus on interculturalism. |
| **Reference** | [31] |
| **Background** | Intervention program aimed at Roma students of Secondary Education. Through action research, the center's needs are identified and a, plan of action for two academic cycles is developed by the educational community leading the group of students in compensatory education. |
| **Objectives** | To design, develop and evaluate an intervention inclusive program for students, teachers and also for the school. |
| **Beneficiary** | Eighteen Roma students and mixing mixed raced students (Spanish, Roma) of a group of compensatory educations, Secondary Education Institute. |
| **Context** | School |
| **School level** | 1st. Secondary Education (compensatory education). |
| **Main results** | The article presents the results of the first year of intervention and improvement proposal for the second. It also presents Increased educational expectations from the tutor. Evidence shows students' better self-esteem, motivation, better grade decrease in disciplinary, action from part of the teachers, greater integration into school activities and decreased absenteeism. |

**Table A1.** *Cont.*

| | |
|---|---|
| **Reference** | [3] |
| **Background** | The Roma Teaching Assistant Program, focuses on an inclusive and multicultural school. Inclusive classrooms were designed, which included aspects of Roma culture that are integrated to teaching methodologies. Roma teaching assistants for Roma children resulted in improved communication amongst all. |
| **Objectives** | To promote inclusion and the adaptation of Roma students to the education system. |
| **Beneficiary** | Fifty students from the ages to 5 and 7 from 9 different classrooms in Latvia. |
| **Context** | School |
| **School level** | Child E. and Primary E. |
| **Main results** | All Roma students were able to successfully integrate the school displaying evident social and academic growth. |
| **Reference** | [32] |
| **Background** | Teaching laboratory experiments is a strategy of learning to increase motivation of Roma students. Assigning an active role to the student, increases their interest and gives control over the learning process. Issues of ecology and environmental protection were selected. |
| **Objectives** | To increase the quality and quantity of knowledge, and motivation of Roma school students. |
| **Beneficiary** | Two hundred and thirty-two students (9–10 years, 21.55% Roma) of primary schools in 4 different cities and villages in Serbia. |
| **Context** | School lab |
| **School level** | 3rd Primary E. |
| **Main results** | Initial tests and final results display an improvement 2.17 times higher than in non-Roma students, thus increasing the ecological laboratory quality, quantity of knowledge, and motivation Roma students. Roma children expressed their satisfaction with the work developed and interest in continuing these projects. |
| **Reference** | [33] |
| **Background** | The intervention is a longitudinal case study of 4 years (2006–2010), which is part of the integrated INCLUD-ED (European Commission VI) EU-funded project. Through the dialogic learning, different activities involving the participation of Roma families in decision-making activities and school to improve children's learning were established. |
| **Objectives** | Implement Successful Educational Actions (SEAs). Establish a dialogic and participatory process among all the educational community. |
| **Beneficiary** | College students and families La Paz, Albacete, Spain. |
| **Context** | School |
| **School level** | Primary E. and Secondary E. |
| **Main results** | The implementation of SEAs has contributed to academic success and reversing situations of inequality. Performance improvements of national standardized tests and increased educational opportunities at school are observed. School absenteeism rates were reduced and an increase in the enrollment percentage of new students was observed. |
| **Reference** | [34] |
| **Background** | Project by the Aristotle University of Thessaloniki and funded by the EU, aimed at young Roma students. A case study related to the importance and use of information technology in delivering new media in the Roma popular culture is presented. |
| **Objectives** | To involve Roma students in the production of knowledge through the use of new technological means. To create a sense of identity and promote their participation through cultural elements. |
| **Beneficiary** | Students in a class support (13 and 15) in a high school in the city of Thessaloniki (Greece). |
| **Context** | School |
| **School level** | Secondary E. |

**Table A1.** *Cont.*

| | |
|---|---|
| **Main results** | Participation and motivation evidenced by students in the different tasks of the project; however, the use of digital technology was not a valid knowledge. The importance of effective methods in teaching Roma students noted. Digital technology helped students overcome Roma obstacles related to written language, therefore, they managed to express themselves more creatively bringing down barriers. |
| **Reference** | [35] |
| **Background** | Programs developed by the Center for Media Research in Education (Eotvos Lorand University, Hungary) affiliated to UNESCO (2005–2013). It was based on collaboration between teachers and families. Developing interdisciplinary programs supported Information Technology and Communication (ICT), trialogical learning theory, art, science and mathematics. They also supported students then in high school. |
| **Objectives** | To improve motivation and to effectively improve social skills, verbal and visual communication as well as promoting earning achievements. |
| **Beneficiary** | Roma multigrade school students from small towns in Hungary. |
| **Context** | multigrade schools |
| **School level** | Primary E. |
| **Main results** | Learning achievements show that, through motivating teaching methods, Roma students are able to learn mathematics and other disciplines. The use of ICT turned out to be a methodology that facilitated access to teaching and learning. |
| **Reference** | [46] |
| **Background** | Education for All (formerly assistant teaching Roma) is one of the major interventions in Southeast Europe since 2009 is administered by the Ministry of Education of Serbia. The strategy of a Roma assistant (teaching assistant) per school is used to support the education of children and strengthen the relationship between family and school. |
| **Objectives** | The objective of this article is to describe general aspects of the program and assess their impact in the first year of implementation. |
| **Beneficiary** | The program is aimed at students Roma primary school students and also Serbian students, as well as families and Roma community. |
| **Context** | School, out of school and community |
| **School level** | Primary E. |
| **Main results** | The results are evaluated in relation to grades in mathematics, Serbian and the number of hours of absence in a year, of Roma children. A greater program impact is evident in schools with fewer Roma children; however, an average positive effect was achieved. Absences were reduced and grades were improved. |
| **Reference** | [40] |
| **Background** | Program to promote school engagement (behavioral and cognitive) of Roma students through storytelling (Sarilhos do Amarelo [41]) |
| **Objectives** | To promote school engagement and to improve participation and motivation. |
| **Beneficiary** | 35 Roma children (10–12 years) two primary schools in Braga, Portugal. |
| **Context** | Extracurricular |
| **School level** | 4th Primary E. |
| **Main results** | Their results highlight the effectiveness of the program to improve school commitment from part of Roma students. Results have shown that with appropriate strategies and methodologies greater school engagement can be achieved in this group. |
| **Reference** | [42] |
| **Background** | Knock, knock It's time to learn! is a 4 year intervention to promote behavioral commitment (truancy and classroom behavior) and school performance of Roma children. For 4 years, the research assistant, according to a structured protocol, knocked on the door of the children in the experimental group to invite them to school. |
| **Objectives** | To increase school attendance, better behavior, performance in mathematics and school progression rate. To inform interventions that have a positive influence on school engagement. |

**Table A1.** *Cont.*

| | |
|---|---|
| **Beneficiary** | Thirty Roma children (6 to 11 years), a city north of Portugal (16 children experimental and 14 control). |
| **Context** | Community |
| **School level** | It begins in year 1st. and ends in year of Primary E. |
| **Main results** | The results of the experimental group demonstrate the effectiveness of intervention in promoting behavioral commitment amongst Roma students. Children improved attendance, behavior, and had higher grades in math, increasing their school progress. The importance of the neighborhood and the social environment awareness, where their culture. Is valued play a vital role. |
| **Reference** | [43] |
| **Background** | Reclaiming Adolescence, the project, developed with three collaborating institutions. They use participatory action-research youth. The project prepared for young Roma and non-Roma, to become researchers. Through training and adequate preparation, young people were involved in the design and implementation of research and the resulting community actions. |
| **Objectives** | Information on educational and professional opportunities of young Roma. To Strengthen the capacity of young researchers to benefit their community. To Develop research strategies following the suggestions of young people. |
| **Beneficiary** | Twenty young people from 15 to 24 years (55% Roma) in Zvezdara and Palilula, Belgrade (Serbia) |
| **Context** | Community |
| **School level** | Secondary E. and University |
| **Main results** | Through the project young researchers were able to develop different skills and knowledge, including leadership, communication, civic, ethnic identity, self-esteem, critical thinking. In addition to a strong commitment to social justice and equity, they also learned about the importance of design education and small-scale projects within their communities. |
| **Reference** | [36] |
| **Background** | Music Workshop was conducted with the collaboration of the University of Uludağ and the Roma Association of Central Bursa. Choral education in the context of music education was developed in order, to strengthen the personality, socialization processes and musical and communicative abilities of Roma students. |
| **Objectives** | To establish a music workshop at school with Roma children |
| **Beneficiary** | Thirty-two Roma children (100% Roma) of a primary school in Mustafakemalpaşa, Bursa (Turkey) |
| **Context** | School |
| **School level** | 4th Primary E. |
| **Main results** | The case study results have shown that music education should occupy an important place in education of Roma students as it plays an important role in their social and cultural background.<br>In addition, an increase in communication musical skills, and peer relationships was seen. A sense of belonging and school motivation, increased school attendance and cultural awareness were also witnessed. |
| **Reference** | [37] |
| **Background** | Increscendo project of intercultural education and collaborative, based on the musical training is based on experiences in similar social contexts. |
| **Objectives** | To generate attitudes and promote intercultural values and personal identity |
| **Beneficiary** | 40 students (58% Roma) School "Antonio Allúe Morer", Valladolid, Spain. |
| **Context** | School |
| **School level** | 2nd–6th E. Primary E. and 1st.Secondary E. |
| **Main results** | The process of social and educational inclusion is encouraged, moving towards intercultural education and development. The results have also identified the development of values, attitudes and knowledge. |

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
