# Peer review of "Intercultural Education for Sustainability in the Educational Interventions Targeting the Roma Student: A Systematic Review"

_sustainability, doi:10.3390/su11123238_

Round 1
Reviewer 1 Report
The manuscript ” A systematic review of intercultural education in the promotion of sustainability by the development of education programs and educational interventions targeting the Roma student” provides a review of 17 articles focusing on educational interventions and programs focusing on improving the inclusion of Roma students. The topic is important and relevant as the drop-out rates of Roma people are notably higher than compared to the rest of the population. The manuscript explains a systematic review of articles focusing on the topic and reports the main findings gained from the 17 articles that are taken for more detailed analysis. The manuscript is written in a clear and detailed way as it describes the main findings from the selected 17 studies in great detail. As a reader I am, however, left with a state of confusion as I am not sure what the main point of this presented analysis is. The authors use a lot of space describing the outcomes of the educational interventions and programs but they do not seem to propose any meta-theoretical discussion about the findings or evaluate the studies based on their choice of approaches or methods. The fact that the findings are presented in a descriptive way makes it difficult to assess the original contribution that the paper aims to make. This may be partly due to that fact that the aim of this paper is not explicitly explained in the text. However, in addition to explaining this more in detail I think the authors should revise the way they present and discuss their findings are currently it seems to be a mere summary of previous papers rather than a thematic discussion or meta-analysis of the original studies.
In order to clarify their message, the authors should revise the introduction and results sections of the paper. One important thing that the authors should especially pay attention to is the way they start the paper currently their manuscript starts by explaining the PRISMA method and only after that they present their topic. The paper also includes several tables labeled as “main aspect of the items considered in the systematic review” but this content is not explained in the text. In the text it is said that “the results are presented in Table 1” but as a reader I am not sure what these “results” are and how the content of the tables represent “the main aspects of items considered in the systematic review”. Table 1 also goes on for several pages and I am not sure what purpose it serves. Please remove the table or include it as an attachment. Please also clarify the relationship between the actual text and the tables as currently they present more or less the same thing.
Author Response
Dear reviewer,
The authors sincerely appreciate your dedication and detailed analysis for this research.
You can find the article with the changes made in attachment. Changes for all reviewers are marked in red.
Next, it is shown the comments made by reviewers and in boxes and red color the changes that have been integrated into the final document in response to each of your suggestions.
REVIEWER 1
I am, however, left with a state of confusion, as I am not sure what the main point of this presented analysis is.
Response:
To answer explicitly incorporated the general and specific objectives of the study, in the introductory section. According to the above, the objective of this study is: to analyze in a descriptive way the interventions directed to the gypsy students in the literature, within the framework of Intercultural Education, and that have been carried out in the educational context. In order to fulfill the aforementioned objective, the following specific objectives have been established:
• Identify the educational context where the intervention has been carried out (school, extracurricular or extracurricular, community or in more than one context). • Describe the main characteristics of the interventions carried out with Roma students according to the purpose of the intervention, school level, participants and results. • Assess the implications of the results of the interventions for Roma students. |
The authors use a lot of space describing the outcomes of the educational interventions and programs but they do not seem to propose any meta-theoretical discussion about the findings or evaluate the studies based on their choice of approaches or methods.
Response:
The results of the descriptive analysis are framed under the general objective of this study, which is performed by a narrative synthesis. The discussion of the findings has been made in paragraph 4. In addition, research methods have not been analyzed for this studys; however, the suggestion will be considered for future research. Also, the table 1 with the outcomes of the educational interventions now is at the annex 1. |
The fact that the findings are presented in a descriptive way makes it difficult to assess the original contribution that the paper aims to make. This may be partly due to that fact that the aim of this paper is not explicitly explained in the text. However, in addition to explaining this more in detail I think the authors should revise the way they present and discuss their findings are currently it seems to be a mere summary of previous papers rather than a thematic discussion or meta-analysis of the original studies.
Response:
The authors acknowledge the comments made, noting that the analysis of the selected items corresponds to a descriptive analysis. (incorporated in the overall objective of the study), at the response 1 we present the new objectives. The suggestion about the findings and the main characteristics that are in line with the elements analyzed as well as the most effective practices has been considered in the discussion and conclusion section of the manuscript. |
One important thing that the authors should especially pay attention to is the way they start the paper currently their manuscript starts by explaining the PRISMA method and only after that they present their topic
Response:
Attention has been paid to how to start writing text, taking into consideration this comment. The manuscript has been modified incorporating the introduction about intercultural education and social inclusion with Roma students is presented At the end of the paragraph the study method is established. |
The paper also includes several tables labelled as “main aspect of the items considered in the systematic review” but this content is not explained in the text. In the text it is said that “the results are presented in Table 1” but as a reader I am not sure what these “results” are and how the content of the tables represent “the main aspects of items considered in the systematic review”. Table 1 also goes on for several pages and I am not sure what purpose it serves. Please remove the table or include it as an attachment.
Response:
Main results are described at the paragraph 3 (Results) and in Table 1 is explained by the text. The table with the basic description about interventions have been included in Annex 1. |

Reviewer 2 Report
The manuscript deals with a timely and relevant issue considering current European recommendations: the integration and academic success of Roma children through intercultural education programs. The authors address this issue by conducting a meta-analysis of European empirical research studies reporting intervention and educational programs that have this goal in mind. Although the study is methodologically sound, based on an existing framework and presented in a detailed manner, from my point of view, it has some limitations that need to be addressed prior to publication:
Theoretical background - key authors in the field of intercultural education are missing (see, inter alia, the work of Michael Byram). Also prior research could be more thoroughly presented, particularly concerning intercultural mediation.
Sustainability - although the promotion of sustainable intervention and inclusion is mentioned both in the title and the introduction, it is regrettably missing from the other sections of the paper. I would suggest coming back to this question in the discussion of the results, for instance by pointing out suggestions/recommendations for similar interventions that promote sustainable change in this area, based on the results of the study.
Presentation and description of the corpus - Table 1 could appear as an appendix at the end of the manuscript. Detailed narration of the empirical studies is enough.
Data analysis and discussion - the manuscript is mainly focused on the description of the corpus/intervention programs, namely the context of the intervention, the target audience, the activities conducted and the results of the research. I feel that a discussion/interpretation of the results of the meta-analysis is missing. This would help the article stand out and become more relevant in the field of intercultural education with Roma children and youth. Questions that could be addressed are, for instance: What are the implications of these results for intervention with this population? What are the implications for research?
Formal aspects - the manuscript presents some linguistic limitations mostly at the syntactic level. Careful English editing is needed, specifically in the sections describing the studies and in Table 1, which has considerable spelling mistakes.
I am attaching the manuscript with further comments. English language mistakes were not corrected.

Author Response
Intercultural education for sustainability in the educational interventions targeting the Roma student: a systematic review
Dear reviewer,
The authors sincerely appreciate your dedication and detailed analysis for this research.
You can find the article with the changes made in attachment. Changes for all reviewers are marked in red.
Next, it is shown the comments made by reviewers and in boxes and red color the changes that have been integrated into the final document in response to each of your suggestions.
REVIEWER 2:
Theoretical background - key authors in the field of intercultural education are missing (see, inter alia, the work of Michael Byram). Also prior research could be more thoroughly presented, particularly concerning intercultural mediation.
A theoretical revision has been included on the main principles of the intercultural education model, including contributions by Michel Byram and other important authors on the subject (lines 91-120), among these are:
1. Byram, M.; Perugini, D. C.; Wagner, M. The development of intercultural citizenship in the elementary school Spanish classroom. Learning Languages 2013, 18, 16–31. 2. Byram, M. Twenty-five years on – from cultural studies to intercultural citizenship. Language, Culture and Curriculum, 2014, 27, 209-225. doi: 10.1080/07908318.2014.974329 3. GUÍA INTER. Una guía para aplicar la educación intercultural en la escuela. Proyecto INTER. Madrid: UNED, 2006. 4. Jordán, J.A. ¿Qué educación intercultural para nuestra escuela? Aula intercultural, 2010, 1-21. 5. Sales, A.; García, R. Educación intercultural y formación de actitudes. Programa pedagógico para desarrollar actitudes interculturales. Revista española de pedagogía 1997, 207, 317-336 6. Jiménez Gámez, R.; Goenechea, C. Educación para una ciudadanía intercultural. Madrid: Síntesis, 2015; pp 1-243. 7. Soriano, A. Convivir entre culturas: Un compromiso educativo. In Educación para la convivencia intercultural; E. Soriano, Eds; La Muralla S.A.: Madrid, España, 2007, pp. 99-126. 8. Muñoz Sedano, A. Enfoques y modelos de educación multicultural e intercultural. In Perspectivas teóricas y metodológicas: lengua de acogida, educación intercultural y contextos inclusivos; Mª. V. Reyzabal, Eds; Dirección General de Promoción Educativa: Madrid, España, 2003, pp. 35-54.
|
Sustainability - although the promotion of sustainable intervention and inclusion is mentioned both in the title and the introduction, it is regrettably missing from the other sections of the paper. I would suggest coming back to this question in the discussion of the results, for instance by pointing out suggestions/recommendations for similar interventions that promote sustainable change in this area, based on the results of the study.
A new discussion and conclusions section has been prepared. It has also been incorporated, based on the results of the study, a series of characteristics and recommendations that should accompany the educational interventions to contribute to an intercultural education for sustainability.
Presentation and description of the corpus - Table 1 could appear as an appendix at the end of the manuscript. Detailed narration of the empirical studies is enough.
It has been considered the suggestion, incorporating Table 1 in annexed.
|
Data analysis and discussion - the manuscript is mainly focused on the description of the corpus/intervention programs, namely the context of the intervention, the target audience, the activities conducted and the results of the research. I feel that a discussion/interpretation of the results of the meta-analysis is missing. This would help the article stand out and become more relevant in the field of intercultural education with Roma children and youth. Questions that could be addressed are, for instance: What are the implications of these results for intervention with this population? What are the implications for research?
The objective of this study has been to analyze, in a descriptive way, the research related to the research questions, therefore, a meta-analysis has not been considered as part of the analysis and discussion; however, it will be taken into account for future research.
To answer the questions raised, recommendations and characteristics have been incorporated to be considered by educators and researchers for the improvement of intercultural education in relation to sustainability in this group. lines 493-525
|
Formal aspects - the manuscript presents some linguistic limitations mostly at the syntactic level. Careful English editing is needed, specifically in the sections describing the studies and in Table 1, which has considerable spelling mistakes.
The authors acknowledge the corrections to the manuscript, which have been considered and valued. Also, an expert native English language has reviewed the full text.
However, we could make use of the journal's linguistic revision services if there are still aspects to improve.
|
About the attached manuscript:
1. I feel that the title is too long and confusing. Please consider revising it to make it more appealing to the reader.
It has been considered the suggestion; therefore, the title of the study has been modified to make it more attractive:
The new title is: “Intercultural education for sustainability in the educational interventions targeting the Roma student: a systematic review”, however, we are open to other suggestions about the title. |
2. It is not clear how this statement fits in the text. "(...) The rates of student dropouts remain high, although they have Decreased, and the difficulty of Promoting the effective transition of Roma students into secondary education or higher education).
The statement: "(...) the rates of student dropouts remain high, although they have Decreased, and the difficulty of Promoting: has been worded differently to give more meaning to the main idea of the paragraph:
For a full and successful social integration of Roma students, educational success is necessary and to achieve this, member countries of the European Union are committed to a set of strategies aimed at inclusive growth, and boosting school attendance rates and the effective transition to higher education.
|
3. You mention only one objective...
The comment is appreciated. To answer explicitly incorporated the general and specific objectives of the study, in the introductory section. According to the above, the objective of this study is: to analyze in a descriptive way the interventions directed to the gypsy students in the literature, within the framework of Intercultural Education, and that have been carried out in the educational context. In order to fulfill the aforementioned objective, the following specific objectives have been established:
• Identify the educational context where the intervention has been carried out (school, extracurricular or extracurricular, community or in more than one context). • Describe the main characteristics of the interventions carried out with Roma students according to the purpose of the intervention, school level, participants and results. • Assess the implications of the results of the interventions for Roma students. |
4. Please consider merging these two questions related to the results. (What have been the results of these educational interventions with Roma students? Does the research show results for the programs or the interventions? Do the intervention programs answer for the political guidelines on educational inclusion of the Roma?
According to the above, they have been reformulated study questions incorporated in the introductory section: (lines 138-142). The investigation’s questions relate to: What programs or interventions were carried out with Roma students? What are the educational levels of the programs or the interventions that have been carried out? What are the settings of these programs or interventions? What have been the results of interventions and their implications for Roma students? Intervention programs respond to the political guidelines on educational inclusion of Roma?
|
5. Please consider placing the tables at the end as an appendix. (Table 1)
Tip considered. Table 1 is included in the annex of the manuscript.
|
6. This statement is unclear. It is worth mentioning that there the process deduplication, which is conducted analysis inclusion criterion 2. Table 3 shows the amount and percentage of duplicates in all databases items in the selection of items.
Table 3 (now Table 2) shows the number of items identified as duplicates, among different databases. It is mentioned that this identification process has been made after the inclusion criterion 2, considering the guidelines of PRISMA. |
7. What you mean is extracurricular, right? (3.2. Studies carried out in the formal context)
It has been considered as "out of the Formal context", interventions that have taken place after school hours, clarified in paragraph 3.2. Studies Carried out in the Formal context.
The clarification, that has been included in the manuscript, is appreciated.
|
8. This study is presented in a more detailed manner than the others. Consistency should be used. [32]
The suggestion has been considered, however, Rosario, et al. (2016) presents an empirical study with a very exhaustive analysis of the variables involved. In any case it has been synthesized to give more consistency to the manuscript (lines 316-325).
|
9. Could this be related the fact that the authors are more aware of studies carried out in their own country? A higher percent of programs or interventions involved not only the Roma community (70.58%) while 29.41% is exclusively aimed at Roma students. Finally, Spain is the country with the most programs or interventions found for this study.
Regarding this observation, the authors have only considered the research found in the existing literature, in no case have they been based on self-knowledge.
This fact is supported by the method used, which follows the guidelines of the PRISMA declaration, as well as the use of advanced searches of each database.
|

Reviewer 3 Report
The paper manuscript consists of a systematic review of intercultural education in analyzing the development of educational programs and interventions addressed to the Roma student. The methodology seems well defined and overall seems an interesting paper.
Below you will find some suggestions for improving it.
Abstract: the abstract has to be focused on the research reporting more details about the results. You can delete the first two sentences which are vague.
Background
Background: Please provide at the beginning more definitions of intercultural education. You can consider also the following papers:
Biasutti, M, Concina E, Frate S. (2019). Working in the classroom with migrant and refugee students: The practices and needs of Italian primary and middle school teachers, Pedagogy, Culture and Society, DOI:10.1080/14681366.2019.1611626
Biasutti, M, Concina E, Frate S. (2019). Social Sustainability and Professional Development: Assessing a Training Course on Intercultural Education for In-Service Teachers, Sustainability, 11 (5), 1238 (ISSN: 2071 - 1050; :Impact factor 2.075). https://doi.org/10.3390/su11051238
- p. 2: Research questions: “What are the educational levels of the programs or the interventions have been carried out?” what do you mean for educational levels? School levels?
- In the research questions is missing a focus on the pedagogical approach and the didactic methods used during the interventions. It could be argued and discussed if a specific pedagogical approach (e.g. learner centered approach) could be more effective or not with Roma students. This could be an important variable to be considered for assessing the efficacy of the educational interventions.
p. 3 please provide more details about the 27 items checklist.
Discussion. The discussion have to be more focused on the critical aspects of previous research identifying the issues and limitations of previous studies. Your analysis could be also an occasion for lunching new trend of research and proposing alternative ways for studying similar issues. Several aspects and ideas could be proposed for developing this field of research and proposing further research.
Limitations: Please add also other limitations of your study
Author Response
Intercultural education for sustainability in the educational interventions targeting the Roma student: a systematic review
Dear reviewer,
The authors sincerely appreciate your dedication and detailed analysis for this research.
You can find the article with the changes made in attachment. Changes for all reviewers are marked in red.
Next, it is shown the comments made by reviewers and in boxes and red color the changes that have been integrated into the final document in response to each of your suggestions.
REVIEWER 3:
1. Abstract: the abstract has to be focused on the research reporting more details about the results. You can delete the first two sentences, which are vague.
It has been considered the suggestion, and there have been some adjustments to the abstract of the manuscript. See the new summary with the changes made, especially lines 25-31 with information about the results. |
2. Background: Please provide at the beginning more definitions of intercultural education. You can consider also the following papers:
Biasutti, M, Concina E, Frate S. (2019). Working in the classroom with migrant and refugee students: The practices and needs of Italian primary and middle school teachers, Pedagogy, Culture and Society, DOI:10.1080/14681366.2019.1611626
Biasutti, M, Concina E, Frate S. (2019). Social Sustainability and Professional Development: Assessing a Training Course on Intercultural Education for In-Service Teachers, Sustainability, 11 (5), 1238 (ISSN: 2071 - 1050; :Impact factor 2.075). https://doi.org/10.3390/su11051238
Both items have been included in the discussion section the cite education and dialogue as a tool in a culturally diverse society, and the importance of training teachers working with disadvantaged groups, promoting intercultural education and social inclusion, as fundamental part of sustainability. |
3. p. 2: Research questions: “What are the educational levels of the programs or the interventions have been carried out?” what do you mean for educational levels? School levels?
The term educational levels (too broad) has been changed by school level
|
4. In the research questions is missing a focus on the pedagogical approach and the didactic methods used during the interventions. It could be argued and discussed if a specific pedagogical approach (e.g. learner centered approach) could be more effective or not with Roma students. This could be an important variable to be considered for assessing the efficacy of the educational interventions.
The authors acknowledge and highly appreciate this comment; however, pedagogical approaches and teaching methods have not been incorporated objective analysis of this study, as well as to the characteristics of each study in question features. We believe that analyzing studies from these approaches would be very significant to evaluate the efficacies of interventions, which will be considered for future research.
However, some pedagogical recommendations have been included based on the characteristics of the successful interventions analyzed.
|
1. p. 3 please provides more details about the 27 items checklist.
In the methodology section it has been specified that the authors have followed the elements of the report (checklist of 27 elements). Due to space limitations, a specification of these has not been included, however they can be consulted in the following reference: Moher, D.; Liberati, A.; Tetzlaff, J.; Altman, DG Preferred reporting items for systematic reviews and meta-analyzes: the PRISMA Statement. PLoS Med 2009, 6, e1000097. doi: 10.1371 / journal.pmed.1000097
|
Discussion. The discussion have to be more focused on the critical aspects of previous research identifying the issues and limitations of previous studies. Your analysis could be also an occasion for lunching new trend of research and proposing alternative ways for studying similar issues. Several aspects and ideas could be proposed for developing this field of research and proposing further research.
The comments and suggestions have been incorporated into paragraph 4 of the manuscript.
More specifically, information has been included on recommendations and characteristics to be considered by educators and researchers for the improvement of intercultural education in relation to sustainability in this colctivo, lines 493-525
|
Limitations: Please add also other limitations of your study.
It has been included as limiting the small number of articles included in this study, which have been very difficult to reach, since the literature often only reports particular educational experiences presented through interviews or qualitative studies, with few studies evaluative programs or educational interventions. Finally, it is considered as a limitation in the search, selection and analysis of research only included articles, not considering other types of documents, which perhaps could have provided any information relevant to the educational intervention towards Roma students. It is specified in section 4.1.
|

Round 2
Reviewer 1 Report
The authors have made significant improvements to their manuscript and as a result of these revisions they address most of the concerns I mentioned in my review. The revised research questions and discussion of the result have clarified the purpose and significance of the study. As a result the focus of the discussion has shifted from being merely descriptive to being analytical and providing the reader with a comprehensive overview of the existing studies. If this is the overall aim of the paper I think this has been reached. The paper could be strengthened even more by pointing out why this type of overview is needed (e.g. is there a lack of knowledge about current studies or is the review beneficial/needed for creating new programs?).
Regarding the purpose of the paper I feel that the authors should still explicitly address how the topic of the paper relates to the journal's theme about sustainability. They address this a little in the discussion section but in the introduction they focus on intercultural education and - even though it can be regarded as an aspect or component of sustainable education - I feel that the authors could explicitly address this in the beginning of their paper. By this I mean that instead of simply stating that sustainability is the goal mentioned in policy document or that intercultural education is a component of sustainable education, I think the authors should address the issues of how and why intercultural education is regarded to be an important element of sustainable education.
There are some spelling issues that should be checked for grammar and clarity. For example, the sentences on p. 3 lines 138-141 and on p. 4 lines 145-147 are almost identical.
Author Response
First of all we would like to thank all the suggestions made that help us improve the quality of the presented article.
Changes since the first version was marked using the red in the text and change control Word.
In relation to what is proposed on:
1. “The paper could be strengthened even more by pointing out why this type of overview is needed (e.g. is there a lack of knowledge about current studies or is the review beneficial/needed for creating new programs?)”.
The suggestion of the reviewer was accepted by introducing a new paragraph between lines 189-195 (new revised version of the manuscript). We hope that we have strengthened the paper.
This descriptive analysis, through a systematic review of previous studies, will identify common and successful aspects (as well as risks and limitations) that, within the framework of intercultural education, present the educational interventions carried out with the group of Roma children for their social and educational inclusion; for this reason it is important to identify, systematize and share the main characteristics, trying to contribute to the generation of useful knowledge for professionals, educators and researchers in this field of intercultural education, a knowledge that allows them to make decisions when carrying out future programs (lines 189-195)
2.- “Regarding the purpose of the paper I feel that the authors should still explicitly address how the topic of the paper relates to the journal's theme about sustainability. They address this a little in the discussion section but in the introduction they focus on intercultural education and - even though it can be regarded as an aspect or component of sustainable education - I feel that the authors could explicitly address this in the beginning of their paper. By this I mean that instead of simply stating that sustainability is the goal mentioned in policy document or that intercultural education is a component of sustainable education, I think the authors should address the issues of how and why intercultural education is regarded to be an important element of sustainable education”.
New Paragraph introduced (Lines 137-144 of the revised manuscript).
Therefore, intercultural education is a valid model to contribute to social and human sustainability; it helps to maintain and increase social cohesion, respect for cultural heritage and guarantee the fulfillment of cultural rights. That is, to obtain sustainable development, intercultural education contributes to the defense of cultural diversity and intercultural miscegenation as the heritage of all humanity and helps to combat all kind of racism and ethnic or social barriers contributing to the equity of genders and races among the people who live in our towns and cities; societies that are increasingly culturally diverse and whose future generations of citizens are the child who today learn to live in our schools.
Finally, an expert who has made some changes to improve these aspects has carried out a revision of the “English language and style”.

Reviewer 2 Report
I would like to thank the authors for making significant changes to their initial version in order to incorporate most of my suggestions. I feel that these have helped to considerably improve the quality of the article.
Still I would like to point out some suggestions in order to strengthen the article and make it more readable:
- explain why your study (a descriptive analysis of empirical research) is relevant to the field of intercultural education;
- check the article again for clarity and spelling.
I am attaching a document with some comments, Editing of English language and style is still a problem. Please consider contacting a professional service.

Author Response
First of all we would like to thank all the suggestions made that help us improve the quality of the presented article.
Changes since the first version was marked using the red in the text and change control Word.
A. Regarding the English language and style, it is suggested that a professional make a linguistic revision of the text. Specifically, two parts that require a thorough review are indicated:
- Lines 98-117
- Table attached.
Both suggestions have been particularly taken into account with the revision of the paragraph (refer to lines 95-136 of the corrected version) and the attached table (see line 763 of the corrected version)
B. In relation to the content:
1.- Line 37. Introduction section. The reviewer says that at the beginning, in the first line, something is missing.
It has been corrected, now it says: Considering the importance of intercultural education for the sustainability [1] and priorities of the European community towards social inclusion [2] (Lines 37-38 of the corrected version)
2.- Suggests varied and specific lexical changes (eliminations, changes of words, introduction of words ...).
The suggested changes are appreciated. They have been accepted in the corrected version of the manuscript.
3.- Line → 96. It is indeed a typo, we change the word "Solidaty" by solidarity ".
It was a typo. It has been corrected.
4.- Line 447 → "Embark" is not appropriate.
Thanks for the suggestion. It has been changed to "cover": (see line 486-487 of the revised manuscript): based on this systematic revision cover educational levels from early childhood up to secondary education.
5. About: “explain why your study (a descriptive analysis of empirical research) is relevant to the field of intercultural education”;
In addition to the relevance and usefulness of the study that were pointed out in the conclusions and discussion section (lines 547-590), the suggestion of the reviewer was accepted by introducing a new paragraph:
This descriptive analysis, through a systematic review of previous studies, will identify common and successful aspects (as well as risks and limitations) that, within the framework of intercultural education, present the educational interventions carried out with the group of Roma children for their social and educational inclusion; for this reason it is important to identify, systematize and share the main characteristics, trying to contribute to the generation of useful knowledge for professionals, educators and researchers in this field of intercultural education, a knowledge that allows them to make decisions when carrying out future programs (lines 189-195 last version).

Reviewer 3 Report
Thank you I am happy with the revisions
Author Response
First of all we would like to thank all the suggestions made that help us improve the quality of the presented article.
Changes since the first version was marked using the red in the text and change control Word.
In relation to what is proposed on “Extensive editing of English language and style required”, an expert who has made some changes to improve these aspects has carried out a revision of the English language and style.
